JGP Journal of General Physiology

Voltage-Gated Na Channels

# Variability in reported midpoints of (in)activation of cardiac $I_{Na}$

Michael Clerx[1,2] , Paul G.A. Volders[2] , and Gary R. Mirams[1]

Electrically active cells like cardiomyocytes show variability in their size, shape, and electrical activity. But should we expect variability in the properties of their ionic currents? In this meta-analysis, we gather and visualize measurements of two important electrophysiological parameters: the midpoints of activation and inactivation of the cardiac fast sodium current, $I_{Na}$. We find a considerable variation in reported mean values between experiments, with a smaller cell-to-cell variation within experiments. We show how the between-experiment variability can be decomposed into a correlated component, affecting both midpoints almost equally, and an uncorrelated component, affecting the midpoints independently, and we find that the correlated component is much larger than the uncorrelated one. We then review biological and methodological issues that might explain the observed variability and attempt to classify each as a within-experiment or a correlated or uncorrelated between-experiment effect. Although the existence of some variability in measurements of ionic currents is well-known, we believe that this is the first work to systematically review it and that the scale of the observed variability is much larger than commonly appreciated, which has implications for modelling and machine-learning as well as experimental design, interpretation, and reporting.

## Introduction

Variability in electrophysiological properties arises at several scales. Between and within subjects, electrically active cells, such as cardiomyocytes and neurons, vary in number (Olivetti et al., 1995), size and shape (Volders et al., 1998), and ion channel expression levels (Schulz et al., 2006). But as we continue down the scales, toward molecules and atoms and into the realms of chemistry and physics, we may expect biological variability to disappear.

Where do ion channels fit in this picture? Transcription, translation, anchoring, and degradation of ion channel proteins can affect the total number of channels in a cell and hence the maximal conductance of its aggregate (whole-cell) currents. But should we also expect cell-to-cell or intersubject differences in properties that are not governed by channel count, such as voltage dependence? Ion channel function is known (or suspected) to be modulated by several mechanisms, including localization, phosphorylation, stretch, and maybe even proximity to other channels (Marionneau and Abriel, 2015; Daimi et al., 2022; Beyder et al., 2010; Hichri et al., 2020). But what is the impact of such mechanisms on variability in "baseline" currents, measured under controlled experimental conditions?

Here, we address this question using literature data gathered for a previous study on the human cardiac fast sodium current, $I_{Na}$ (Clerx et al., 2018). Where our earlier study focused on mutants, here we shall use exclusively the accompanying wild-type controls. To gain a large but uniform data set, we will focus on the most common experiment type in this database: measurements using the whole-cell patch-clamp configuration in cells heterologously expressing *SCN5A*, the primary subunit of the channels conducting $I_{Na}$ in the human heart. Although $I_{Na}$ voltage dependence is complex, we shall focus on two of the most common quantities used to characterize it: the midpoints of activation ($V_a$) and inactivation ($V_i$). These describe the voltage at which the channel is half-maximally activated (or the voltage at which the measured peak conductance is half the maximum observed value) and the voltage at which it is half-maximally inactivated (see e.g., Sakakibara et al., 1992; Chadda et al., 2017).

In the Background section below, we introduce the type of experiment and analysis performed in the studies we surveyed.

[1]Centre for Mathematical Medicine and Biology, School of Mathematical Sciences, University of Nottingham, Nottingham, UK; [2]Department of Cardiology, Cardiovascular Research Institute Maastricht, Maastricht University Medical Center, Maastricht, Netherlands.

Correspondence to Michael Clerx: michael.clerx@nottingham.ac.uk

This work is part of a special issue on Voltage-Gated Sodium (Na_v) Channels.

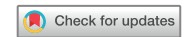

Figure 1.   **Cardiac I_Na and measurement protocols. (A)** A diagram of the ventricular action potential, the transient I_Na during an AP (with an inset showing the first few ms during which the current activates and inactivates), and the late I_Na during an AP. **(B)** A schematic illustration of an activation protocol, the resulting currents with peaks indicated by circles, and the peak currents plotted against the test step voltage. **(C)** A schematic of an inactivation protocol, the resulting currents with peaks indicated by triangles, and the activation (green circles) and inactivation (purple triangles) curves derived from the peak currents. Numbers and shapes are chosen similar to András et al. (2021) for panel A and Nagatomo et al. (1998) for panels B and C. AP, action potential.

Those familiar with activation and inactivation experiments may wish to jump ahead to Materials and methods or Results.

### Background

In healthy cardiomyocytes, I_Na is a brief inward current of a very large magnitude (Fig. 1 A) that powers the initial upstroke of the action potential in response to stimuli from neighboring cells. As such, it is a principal determinant of cardiac conduction velocity, and reduction of I_Na is associated with conduction disorders and risk of reentrant arrhythmias (King et al., 2013). This transient I_Na during the upstroke is followed by a much smaller late (sustained/persistent) component present throughout the action potential, which, if increased, can lead to early afterdepolarizations, long-QT syndrome, and related disorders (Horváth et al., 2022). Recovery of sodium channels upon repolarization contributes to the refractory period, and extraction of the Na+ carried in by I_Na is a major part of ionic homeostasis.

Central to I_Na kinetics are the processes of activation, by which channels open, and inactivation, by which opened channels are blocked. Upon repolarization, the channels deactivate (reverse activation) before recovering from inactivation (Kuo and Bean, 1994). The voltage dependence of activation and inactivation is commonly investigated by applying voltage-step protocols like those shown in Fig. 1. The activation protocol (Fig. 1 B) consists of long periods at the holding potential to let channels deactivate and recover, before brief steps to an incremental test potential are applied. As the test potential is increased, the current appears at around −60 mV and then grows in magnitude, reaching a peak near −20 mV (under physiological conditions). The protocol for inactivation is similar to that for activation, except now the incrementing voltage step is used as a

preconditioning step and followed by a test step at a fixed potential (e.g., −20 mV). During the preconditioning step, a fraction of the channels inactivate, and this is reflected in the current measured during the test step, which decreases when the preconditioning potential is raised.

To analyze the experiments, the peak current during each test step is measured and plotted against either the test potential (activation) or the preconditioning potential (inactivation). A curve is then fit by assuming that the current is ohmic, that the inactivation during the activation experiment (and vice versa) contributes a constant factor that can be cancelled out through normalization, and that the voltage dependence of the peak current in either experiment is due to a single rate-limiting transition, which can be described by a Boltzmann distribution (Hanck and Sheets, 1992; Hille, 2001). Under these assumptions, the peak currents during the activation process can be fit by

$$I_{peak,a} = \frac{g_{max,a}(V_{test} - E)}{1 + e^{(V_{test} - V_a)/k_a}},$$

where $V_{test}$ is the test potential, $E$ is the reversal potential (measured experimentally or calculated from the Nernst equation), and $V_a$, $k_a$, and $g_{max,a}$ are obtained through curve fitting. The equation for the activation curve is found by omitting the normalization factor $g_{max,a}$ and the ohmic-driving term $(V_{test} - E)$

$$\frac{1}{1 + e^{(V_{test} - V_a)/k_a}}.$$

Here, $V_a$ is the midpoint of activation, and $k_a$ determines the curve's slope—with the sign convention used here, $k_a$ is a positive number, and a smaller value indicates a steeper slope. For inactivation, which takes place during the preconditioning step but is measured in the test step, the equation becomes

$$I_{\text{peak,i}} = \frac{g_{\text{max,i}}(V_{\text{test}} - E)}{1 + e^{(V_{\text{pre}} - V_i)/k_i}} = \frac{I_{\text{max}}}{1 + e^{(V_{\text{pre}} - V_i)/k_i}},$$

where $I_{\text{max}}$ is the largest (most negative) current measured during the protocol. Again, we can omit the numerator to find the inactivation curve, with midpoint $V_i$ and a slope determined by $k_i$—with this sign convention, $k_i$ is a negative number. Example activation and inactivation curves and their midpoints are shown in Fig. 1 C.

The procedure is then repeated for multiple cells, and values for $V_a$ and $V_i$ are averaged to obtain the mean midpoints $\mu_a$ and $\mu_i$, along with an estimate of the standard deviation (or more commonly the SEM) in either quantity. For this study, we collected these $\mu_a$ and $\mu_i$ from several published works but did not perform any new experiments or experimental analysis.

Further background on $I_{\text{Na}}$ is given in, e.g., Chadda et al. (2017), Armstrong and Hollingworth (2021), Amin et al. (2010), Patlak (1991), Catterall (2012).

## Materials and methods

All data used in this study were collected as part of a previous study on single-point mutations in *SCN5A* in expression systems (Clerx et al., 2018). For the current study, we reduced this data set to keep only wild-type (control) measurements, we removed *Xenopus* oocyte measurements to keep only whole-cell patch-clamp studies, and we added additional metadata as detailed below. The systematic process whereby the original and novel data were gathered is detailed below. Although this is not a study into effect sizes, we followed the PRISMA guidelines (Page et al., 2021) where applicable.

To identify candidate studies, we searched PubMed for "SCN5A mutation" (with the last search occurring in May 2016) and looked in previously published lists of mutations (Napolitano et al., 2003; Moric et al., 2003; Ackerman et al., 2004; Zimmer and Surber, 2008; Hedley et al., 2009; Kapplinger et al., 2010; Kapplinger et al., 2015). Studies identified this way were then scanned to see if they contained measurements of $V_a$ or $V_i$ made with whole-cell patch clamp in either HEK or CHO cells, along with the number of cells measured and a standard deviation or SEM. Next, we filtered out studies made at normal or raised body temperatures but kept studies made at "room temperature" (as stated by the authors) or at any temperature in the range from 18 to 26°C. Because of the considerable effort involved in performing experiments at body temperature, we assumed that studies not mentioning temperature satisfied our criteria and could be included. Similarly, we excluded any studies under non-baseline conditions (e.g., with known stretch, remodelling, ischemia, etc.). All data collection and selection was performed by M. Clerx.

The dataset includes measurements in two different expression systems: HEK293 or tsA201 (both indicated as "HEK" in this study) and CHO cells. A clear statement of cell type was part of the inclusion criteria (see above) so that no missing-data strategy was required. The exact *SCN5A* α-subunit expressed in these cells was not always clearly indicated. We found at least four different isoforms, which we labelled: a, sometimes known

as Q1077 and with GenBank accession number AC137587; b, known as Q1077del or GenBank AY148488; a*, hH1, R1027Q, or GenBank M77235; and b*, hH1a or T559A; Q1077del, no GenBank number (see also Makielski et al., 2003). Missing α-subunit information was recorded as "α-subunit unknown." Finally, we noted whether or not studies stated a co-expressed β1 subunit; no information on β1 co-expression was taken to mean it was not co-expressed.

Some studies we surveyed recorded separate control (wild-type) experiments for each mutant (see Table S1). We therefore distinguish between studies and experiments, where a study can contain several experiments, and each experiment summarizes findings in multiple cells.

For each experiment, we noted either the midpoint of activation (as a mean $\mu_a$, a sample standard deviation $\sigma_a$, and a cell count $n_a$), the midpoint of inactivation ($\mu_i$, $\sigma_i$, and $n_i$), or both. All numbers were taken from publications at face value: no new curve fitting or other analysis of experimental traces was performed for this study. Sample standard deviations were not usually provided in publications but could be calculated from the provided SEMs. In Fig. 2, we shall make the further assumption that midpoints in individual cells were distributed normally, allowing us to plot a 5th-to-95th percentile range of the corresponding normal distribution.

Where both midpoints were reported, cell counts were often equal (34%) or similar (differing by no >5 cells in 90% of experiments; see Table S2). So while it is plausible that $V_a$ and $V_i$ were often both measured in the same cell, this cannot be guaranteed (and was not explicitly stated in many papers). However, we will assume that, even when cell counts were different, the conditions under which $V_a$ and $V_i$ were measured in an experiment were similar enough that correlations between $\mu_a$ and $\mu_i$ can be studied.

In a second pass over the selected papers, performed between September 2024 and May 2025 by M. Clerx, additional metadata were gathered on the experiments, including whether liquid junction potential (LJP) correction was performed, the voltage at which maximal $I_{\text{Na}}$ was measured in the activation protocol, the slope of the activation and inactivation curves, the voltages used in the activation and inactivation protocols, and the bath and pipette solutions.

### Online supplemental material

The supplemental materials contain an extended version of Fig. 5 (Fig. S1), a table listing the studies that contained multiple experiments (Table S1), a table listing the number of cells used in experiments (Table S2), and a full list of all the included midpoints, with standard deviations, cell counts, and literature references (Table S3). Fig. S1 shows the correlations between reported experimental factors and mean midpoints of activation ($\mu_a$) and inactivation ($\mu_i$), indicated by an orange linear regression line.

## Results

In the 117 studies that met the selection criteria, we found a total of 172 experiments: 150 experiments reporting both midpoints,

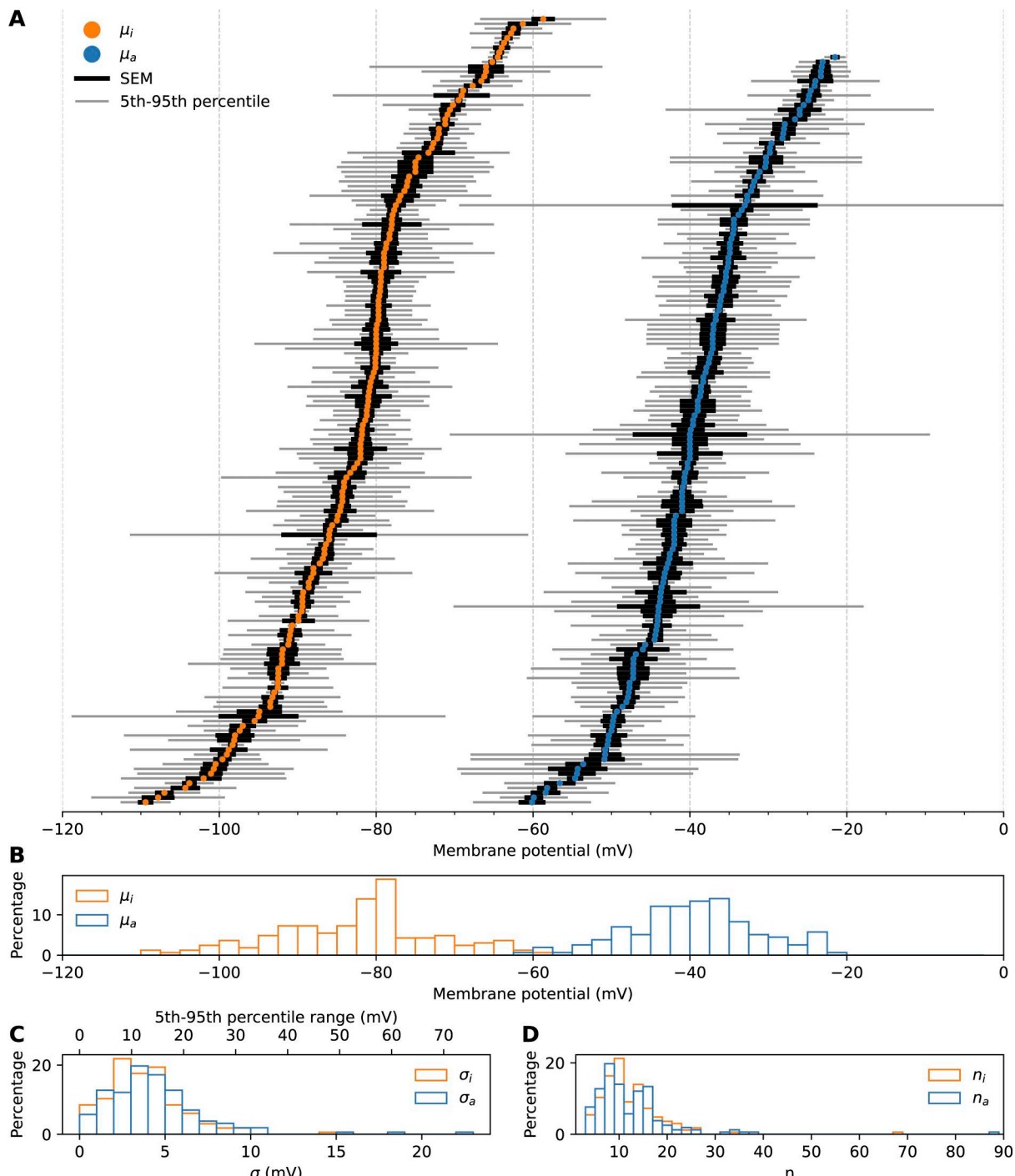

Figure 2. **Variability in mean midpoints of I_Na. (A)** Reported mean midpoints of inactivation ($\mu_i$, left) and activation ($\mu_a$, right) for all experiments. Vertically, both sets of points are individually ordered from most to least negative membrane potential: correlations between an experiment's $\mu_a$ and $\mu_i$ cannot be seen here and will be examined in Fig. 3. The SEM for each experiment is indicated by a thick black bar. A thinner grey bar shows the 5th-to-95th percentile range of a normal distribution with the reported mean and standard deviation: if the individual cell measurements in these studies were normally distributed, 90% of measurements would fall within this range. **(B)** A histogram view of the means. The y axis shows the percentage of reported means with each potential. **(C)** A histogram view of the standard deviations. A second x axis (top) shows the corresponding 5th-to-95th percentile ranges. **(D)** A histogram view of the number of cells measured per experiment.

7 reporting only on activation, and 15 reporting only on inactivation. Each experiment in our data set consists of measurements of $V_a$ and/or $V_i$ in several cells, reported as a mean ($\mu_a$ and $\mu_i$, respectively), a standard deviation ($\sigma_a$ and $\sigma_i$), and a cell count ($n_a$ and $n_i$). The obtained means ($\mu_a$ and $\mu_i$) and SEM are shown graphically in Fig. 2. To see where the individual cell estimates of $V_a$ and $V_i$ may have been, for each experiment, we also plot the 5th-to-95th percentile range of a normal

distribution with the reported μ and σ (approximately the range μ ± 1.64σ). We shall use the individual standard deviations as a measure of within-experiment variability and refer to the difference between the means as between-experiment variability.

Within-experiment variability can be seen in the grey bars in Fig. 2 A and the histograms in Fig. 2 C. The median standard deviations were 3.6 mV for $\sigma_i$ and 4.0 mV for $\sigma_a$. Assuming a normal distribution, this suggests that 90% of single-cell results in a typical experiment fall in a range of ~12 mV ($V_i$) to 13 mV ($V_a$). Slightly larger ranges of up to 20 or 30 mV are also not uncommon (Fig. 2 C, top axis), and outliers go up to 50 mV ($V_i$) and 73 mV ($V_a$).

More surprisingly, substantial between-experiment variability can be seen in Fig. 2, A and B: reported means $\mu_i$ range from –109 to –59 mV (median –81.2 mV, range 50.7 mV, 5th-to-95th percentile range 35.7 mV), while the means $\mu_a$ range from –60 to –21 mV (median –39.9 mV, range 38.6 mV, 5th-to-95th percentile range 29.0 mV). Despite the large between-experiment variability, the SEM for most experiments, which quantifies the degree of certainty in the estimate of the mean, is quite narrow. This suggests that the mean $V_a$ and $V_i$ differed significantly between the surveyed experiments and that one or more confounding factors may exist that explain this difference. Inactivation results seem more affected, with a much larger between-experiment variability for $\mu_i$, while the median within-experiment variability $\sigma_i$ is slightly smaller than $\sigma_a$.

Cell counts per experiment are shown in Fig. 2 D and ranged from 3 to 88 (activation) and 3 to 68 (inactivation), with a median of 10 for both $n_a$ and $n_i$.

## Mean midpoints $\mu_a$ and $\mu_i$ strongly correlate across experiments

Next, we look at $\mu_a$ and $\mu_i$ in the subgroup of 150 experiments where both were reported, as shown in Fig. 3 A. Each experiment is indicated by a dot, and a linear fit through all experimental means is shown, made using unweighted least squares based linear regression. This line had an offset of –45.7 mV and a slope of 0.93 mV/mV, with a Pearson correlation coefficient of r = 0.79. The coefficient of determination was $r^2 = 0.62$, indicating that 62% of the variance is explained by this linear correlation. A second regression with a fixed slope of 1 is shown (green line), and this falls within the 95% confidence interval of the original regression (shaded grey area and dashed blue lines), so that we cannot statistically reject the hypothesis that the slope equals 1. Together, this correlation suggests the existence of some unknown factors shifting $\mu_a$ and $\mu_i$ by approximately equal amounts between experiments.

We can decompose the difference between each ($\mu_a$, $\mu_i$) measurement and the group mean into a component along the line of best fit (without constraining the slope) and a component perpendicular to the line of best fit (i.e., principal component analysis). An example for a single point is shown by the arrows drawn in Fig. 3 A), and the same example point is highlighted in black in panels B and C. The result suggests that most of the between-experiment variability is positively correlated.

In panels B and C, we test whether the variability in either direction diminishes with experiment size (number of cells

tested). To this end, we define "experiment size" as the number of cells $n_i$ tested to measure $\mu_i$, plus the number of cells $n_a$ tested to measure $\mu_a$. In Fig. 3 B, we plot the square root of this quantity ($\sqrt{n_a + n_i}$) as a function of the first principal component to create something akin to a "funnel plot." No clear triangle shape is observed in either plot, but the first component does appear to diminish somewhat for the experiments with an increased number of measurements.

## Subunits and cell type are not the major sources of variability

Cell type, α-subunit isoform, and β1-subunit co-expression may affect $V_a$ and $V_i$ and were duly reported in most publications we checked. But can they explain the large between-experiment variability we observed? In Fig. 4, we show the same data as in Fig. 3, but grouped by recorded α-subunit, β1 co-expression, and cell type. The largest subgroup (a* subunit, with β1 co-expression, in HEK) is shown in Fig. 4 D. It is clear that, while some differences between these groups exist that could cause subtle shifts in the means, grouping like this does not divide our data into clear-cut clusters. In fact, many of the larger groups span the full observed range, suggesting that these factors have only a small effect on $V_a$ and $V_i$ measurements—even though their effect on in vivo electrophysiology may be profound.

## Within-study between-experiment variability

The last two panels in Fig. 4 show the two studies with more than five experiments: Kapplinger et al. (2015) (27 experiments) and Tan et al. (2005) (15 experiments). Again, a strong correlated component is visible in both. Compared with the full data set, both correlated and uncorrelated between-experiment variability are much smaller in these groups.

## Experimental variability

Uncorrected LJPs are possible confounders causing an equal change in $\mu_a$ and $\mu_i$ (see Discussion). Only 10 studies surveyed mentioned correcting for the LJP, 7 mentioned not correcting, and the remainder did not report on LJP correction. Fig. 5 A compares known corrected and known uncorrected experiments (where these included both $\mu_a$ and $\mu_i$).

Holding potentials and times can influence the measured midpoints, and of these, the potentials were more regularly reported. Fig. 5 shows the correlation between the holding potential in the activation experiment, $V_{hold,a}$, and $\mu_a$ in Fig. 5 B, and between $V_{hold,i}$ and $\mu_i$ in Fig. 5 C. Here, we find coefficients of determination of 0.29 and 0.15, respectively. While this shows that 29% of the variability in $\mu_a$ can be predicted if the chosen $V_{hold,a}$ is known, it should be stressed that the relationship is not necessarily causative: studies typically copy several design aspects from predecessors, so that the differences could be due to shared confounding variables.

Loss of voltage clamp in the activation experiment can cause an increased steepness of the activation curve, $k_a$, and a leftward (hyperpolarizing) shift of $V_a$. In Fig. 5 D, we plot $\mu_a$ against $k_a$ (where reported), but no correlation is observed (see Discussion for a possible explanation).

Finally, the voltage at which peak current occurs during the activation experiment ($V_{peak}$) depends on both $V_a$ and $V_i$. In

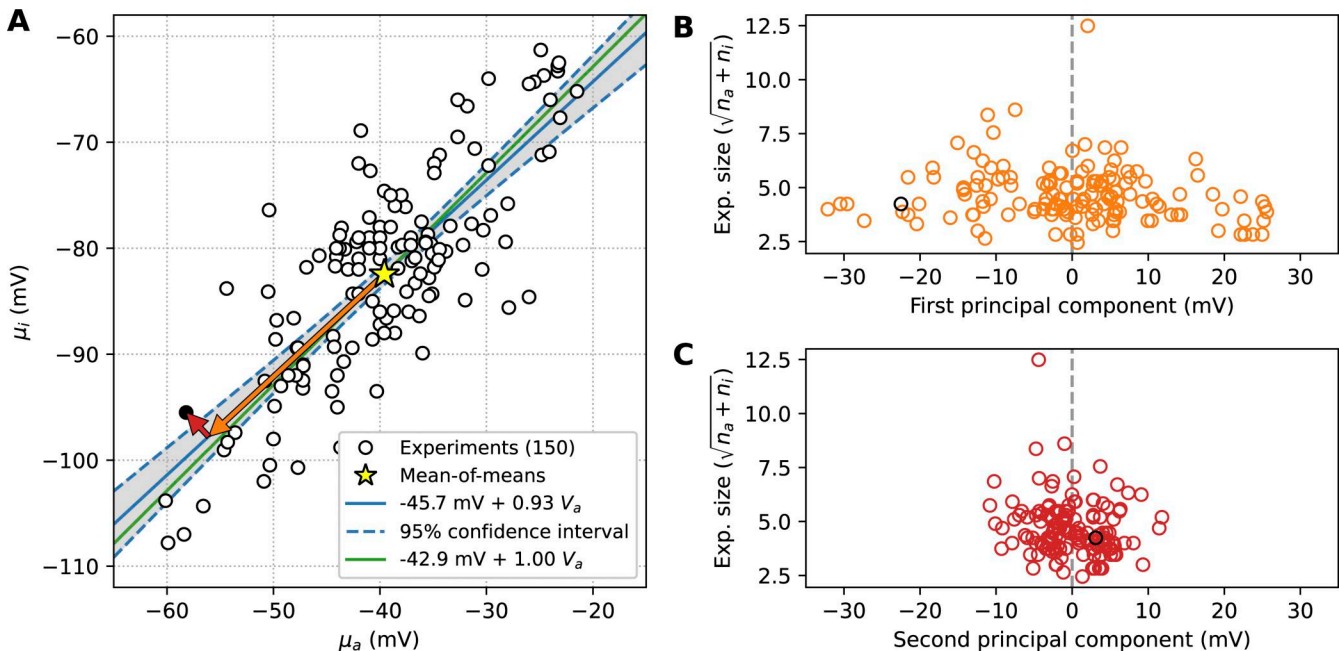

Figure 3. **The midpoints are strongly correlated, and variability can be decomposed into a correlated and uncorrelated component. (A)** Mean midpoints of inactivation $\mu_i$ plotted against mean midpoints of activation $\mu_a$ for the 150 experiments that reported both. The mean of all points (a mean-of-means) is indicated by a yellow star. A best-fit line is shown as a solid blue line, with its 95% confidence interval indicated by dashed blue lines and a grey shaded area. A second linear regression line with a slope constrained to have a gradient of one is shown in green. For one example experiment ($\mu_a$ = −58.2 mV, $\mu_i$ = −95.5 mV), we show the vector from the mean-of-means to this point, decomposed into components along the line of best fit (orange, first principal component) and perpendicular to the line of best fit (red, second principal component). The same example point is highlighted in black in panels B and C. **(B)** The square root of the experiment size as a function of the first principal component, for all points in A. The experiment size is defined as $n_a + n_i$, where $n_a$ is the number of cells tested for $V_a$ and $n_i$ is the number tested for $V_i$. **(C)** The square root of the experiment size as a function of the second principal component.

Fig. 5 E, we show a strong relationship between $V_{peak}$ and $\mu_a$, providing further evidence that the midpoints are correlated.

## Discussion

We observed strong variability within experiments (median $\sigma_i$ was 3.6 mV, median $\sigma_a$ was 4.0 mV, but with outliers up to 22 mV) and between experiments ($\mu_i$ ranged from −109 to −59 mV, $\mu_a$ from −60 to −21 mV) and found a strong positive correlation across experiments measuring both (explaining 62% of the observed between-experiment variance). Cell type, α-subunit, and β1-subunit were seen to have an influence, but grouping by these categories did not explain the results. We also saw within-study between-experiment variability on a smaller scale but with a visually similar correlation. How should we interpret these findings?

The existence of *some* within-experiment variability is well known and is the reason why midpoints are reported as a mean and SEM. The existence of between-study or between-lab variability, too, is indirectly acknowledged by the mutant studies we collected in Clerx et al. (2018) and reused here: each provided a new wild-type recording instead of using a value from the literature. Some studies measuring multiple mutants have gone even further and accounted for within-study between-experiment variability by performing a paired control wild-type measurement for every measured mutant. For good examples, see Kapplinger et al. (2015) (27 reported wild-type

values) or Tan et al. (2005) (15 reported wild-type values). The Tan et al, 2005 paper also provides the only direct acknowledgment of between-experiment variability we found, citing "seasonal variation in current characteristics" as a reason for their paired study design. However, the wild-type values reported in Tan et al. (2005) and Kapplinger et al. (2015) differ by at most 11 and 7 mV, respectively—well short of the 40 and 50 mV ranges seen in Fig. 2. The full extent of between-experiment variability, then, is still surprising.

Interestingly, the least negative (most depolarized) reported value of $\mu_i$ is −58.7 mV, exceeding the most negative (least depolarized) $\mu_a$ of −60.1 mV. Such a situation is clearly not physiological, and it is tempting to postulate some unknown biological mechanism (present even in cells non-natively expressing *SCN5A*) that regulates the difference between the midpoints, keeping $V_a$- $V_i$ at ~45 mV and explaining the correlation with a gradient indistinguishable from 1 that is seen in Fig. 3. However, a simpler explanation might be sought in experimental factors causing a difference between the intended and the applied voltage that applies equally to measurements of $V_a$ and $V_i$. We briefly review possible factors below.

### Experimental sources of variability

An overview of experimental sources of variability (or more precisely, *uncertainty* that might cause variability in measurements; see Mirams et al. [2016]) is shown in Table 1, and we have made a tentative effort to classify each as causing between- or

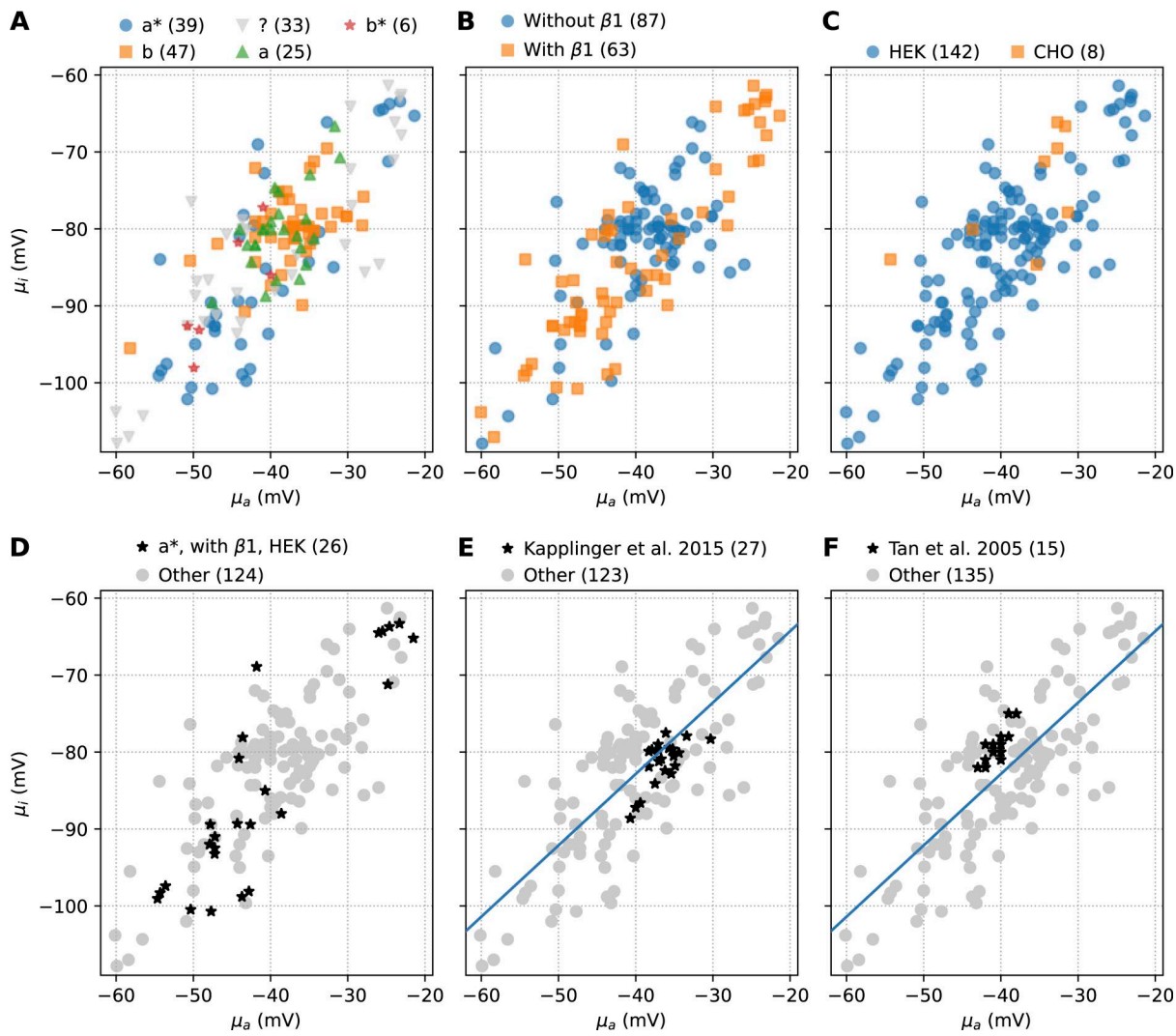

Figure 4. **Grouping by recorded α-subunit, β1 subunit co-expression, and cell type does not create distinct clusters and only explains a small part of the observed between-experiment variability.** The number after each category indicates the corresponding number of means. Within-study between-experiment variability is observed in the two largest studies but is much smaller than in the full data set. **(A)** Grouping by α-subunit: from largest to smallest subgroup, we show the a* (R1027Q) α-subunit, b (Q1077del), not reported, a (Q1077), and b* (T559A; Q1077del). **(B)** Grouping by β1 co-expression. **(C)** Grouping by cell type (HEK versus CHO cells), but note the very different group sizes. **(D)** The largest subgroup versus all other results. **(E and F)** Within-study variability in the works by Kapplinger et al. (2015) and F, Tan et al. (2005). The blue line in E and F is the linear regression line to the full data set shown in Fig. 3.

within-experiment variability. The between-experiment column is further divided into correlated and uncorrelated effects. Disputed or hypothetical factors are indicated using question marks, while check marks indicate factors known to strongly influence results—although the extent of their effect on our data is still unknown. In the text below, we explain our reasoning and, where possible, provide speculative upper bounds on effect magnitudes.

### LJP

LJPs need to be considered when a liquid–liquid interface changes after the recorded current has been "zeroed" during a voltage-clamp experiment (e.g., by breaking the seal), and they are usually corrected by applying a calculated voltage offset. Typical LJP values in patch-clamp electrophysiology have been estimated as 2–12 mV (Neher, 1992). Different values are

expected in different experiments, as different bath and pipette solutions are used. An appropriate correction would be expected to remove variation completely by providing the appropriate membrane voltage regardless of solutions. But failure to correct, a systematic error in the correction or, in the worst case, a sign error in the correction could lead to equal errors in both midpoints of up to 24 mV. In addition, the exact LJP correction is difficult to calculate, and depends on chelating agents, pH buffers, any NaOH, CsOH, or other salt added to adjust the pH, and even the LJP calculation method (Marino et al., 2014, Preprint), so that a few mV variations even between LJP-corrected data are expected. Only 10 out of 117 studies surveyed (8.5%) stated LJP correction was applied, with corrections ranging from 6.7 to 8 mV. Fig. 5 A shows that the LJP-corrected studies are all at the lower range of reported values, while known uncorrected studies occupy the upper half. However, as with panels B and C

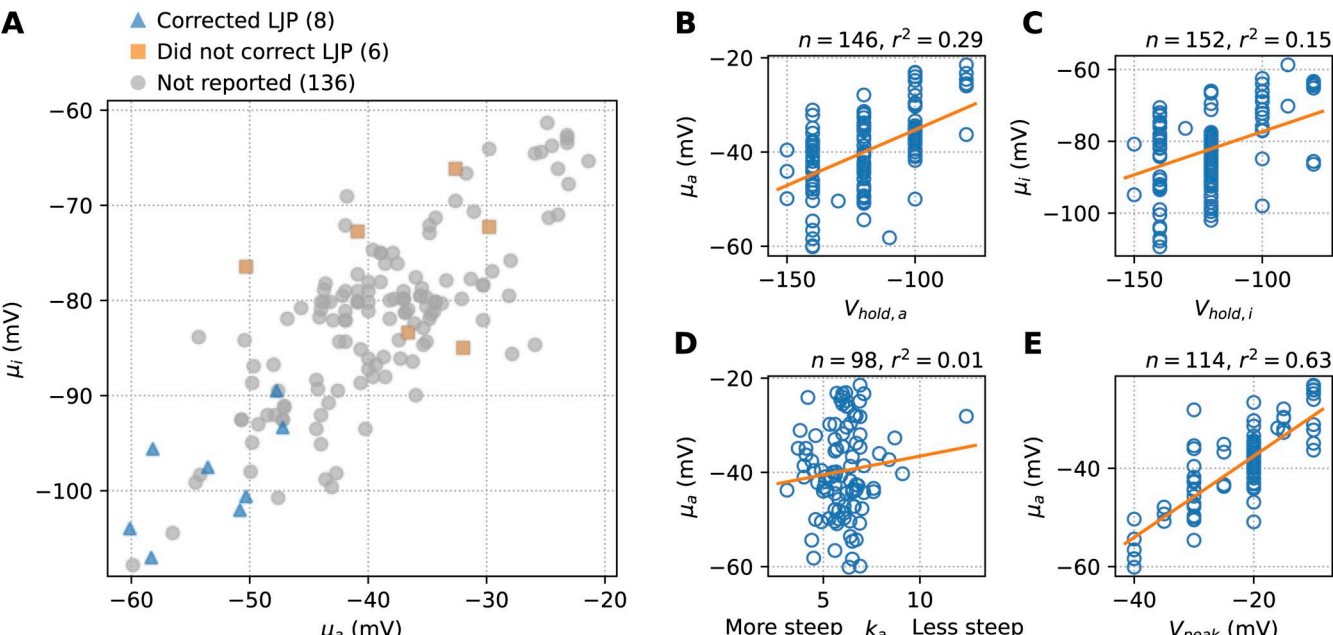

Figure 5. **Correlations with experimental factors. (A)** Mean midpoint of activation, $\mu_a$, and inactivation, $\mu_i$, in studies that report correcting for the LJP, that report not correcting, and that do not report on LJP correction. **(B and C)** $\mu_a$ and $\mu_i$ versus the holding potential in the activation and inactivation protocols, respectively. A regression line is shown, along with the number of studies for which this data were available, n, and the coefficient of determination, $r^2$. **(D)** We saw no correlation between $\mu_a$ and the steepness of the activation curve, $k_a$. **(E)** The voltage at which the peak current occurred during the activation protocol correlates strongly with the reported $\mu_a$.

in this figure, it is possible that other shared design choices caused or contributed to this effect.

### Redox potentials

A related possible cause of variation during an experiment is if the electrode potential changes after zeroing. This is typically encountered (and noticed) when electrode chlorination levels are low, but drift on slower time scales (e.g., 10 min) could easily go unnoticed, causing some within-experiment variability. The size of the effect depends on pH and chloride concentration (Berman and Awayda, 2013), but we saw no direct correlation with chloride concentrations in Fig. S1.

### Voltage control errors

$I_{Na}$ is characterized by fast time scales and large current amplitudes, both of which cause problems for membrane potential control in voltage-clamp experiments (Sherman et al., 1999; Lei et al., 2020; Montnach et al., 2021). In particular, a combination of cell capacitance (which increases with size) and series resistance can cause large shifts in either midpoint. Techniques such as series resistance compensation are commonly used, but even then shifts as large as 10 mV can be incurred (Montnach et al., 2021), while under less favorable conditions shifts of 20 mV (Montnach et al., 2021) or 30 mV (Abrasheva et al., 2024) can be expected. Because the size of this effect depends on cell size and the achieved series resistance, we can expect variability within experiments, and because it depends on quality control procedures and the precise technology used in the lab, we can also expect (correlated and uncorrelated) between-experiment effects, so that we classify voltage control errors as contributing to

all three columns of Table 1. Loss of control in the activation protocol can sometimes be detected from depolarizing shift in $V_a$ and an increased steepness (smaller $k_a$) of the activation curve (Montnach et al., 2021; Abrasheva et al., 2024). However, this effect can be hidden by averaging over multiple cells (Lei et al., 2025), which may explain the lack of correlation between $\mu_a$ and $k_a$ in Fig. 5 D.

### Voltage protocol

Voltage step protocols vary between studies and can affect the results. For midpoints, which are steady-state properties, a major factor will be the duration of the steps intended to bring the channel into steady state (for an example in $I_{Kr}$, see Vandenberg et al. [2012]). Similarly, the choice of holding potential will affect the rate at which channels transition, making this another important parameter. Although holding times were not well reported in the surveyed data, we do present a plot of holding potential and its correlation with $\mu_a$ and $\mu_i$ in Fig. 5, B and C. Assuming this effect depends only on the experimental approach and not on the individual cells, we assign it to both between-experiment columns of Table 1.

### Analysis method

Several methods exist to filter current data, extract peaks, fit curves and/or normalize the data. Though the size and direction of such effects is hard to predict, the choice of method varying between studies can cause (most likely uncorrelated) between-experiment variability, while the reliability of the method (particularly sensitivity to noise) can lead to within-experiment variability.

Table 1. **Postulated experimental causes of variability, grouped as correlated between-experiment (affecting $\mu_a$ and $\mu_i$ similarly in each experiment), uncorrelated between-experiment (affecting $\mu_a$ and $\mu_i$ independently in each experiment), or within-experiment (affecting $\sigma_a$ and/or $\sigma_i$)**

| | Between experiment | | Within experiment |
| --- | --- | --- | --- |
| | Correlated | Uncorrelated | |
| Missing or erroneous LJP correction | ✓✓ | ✗ | ✗ |
| Uncontrollable redox potentials | ? | ? | ✓ |
| Voltage-control errors | ✓ | ✓ | ✓ |
| Voltage protocol | ✓ | ✓ | ✗ |
| Analysis method | ✗ | ✓ | ✓ |
| Bath and pipette solutions | ✗ | ✓ | ? |
| Temperature | ? | ✗ | ? |
| Time since rupture | ? | ? | ? |
| Stretch | ✗ | ✗ | ? |
| Culture conditions and passage number | ? | ? | ? |
| Endogenous currents | ? | ? | ? |
| Regulation | ? | ? | ? |

Characterized as strongly likely (✓✓), likely (✓), possible (?), or unlikely (✗) to contribute to the different types of variability in measurements of $\mu_a$ and $\mu_i$.

### Bath and pipette solutions

The exact compositions of bath and pipette solutions (including buffers, chelating agents, and blockers for endogenous currents) could affect the results. For example, high concentrations of calcium in the pipette are known to induce a depolarizing shift in $V_i$ (Van Petegem et al., 2012). In Fig. S1 we show that, in our data set, no strong linear correlations could be seen between the midpoints and sodium, calcium, and chloride concentrations. Nevertheless, we include the solutions as a possible source of uncorrelated between-experiment variability.

### Temperature

The measurements we reviewed were made at room temperature, defined by the various authors as anywhere between 18 and 26°C. Nagatomo et al. (1998) recorded a shift in the midpoint of activation of +0.43 mV per °C and a +0.47 mV per °C shift for inactivation, although no such shifts were observed by Keller et al. (2005), and both studies used HEK cells. If there is a 0.5 mV per °C shift, the observed range of room temperatures could lead to a correlated between-experiment effect of up to 4 mV. Within studies, temperature was usually given as a 1 or 2° bracket, leading to a much smaller within-experiment estimate of ±0.5–1 mV.

### Time since rupture

Hanck and Sheets (1992) measured $I_{Na}$ in Purkinje cells and studied the effect of the time between rupturing the membrane and performing the measurement, which caused both midpoints to drift toward more negative potentials at ∼0.5 mV per minute. A study by Abriel et al. (2001) looked for, but did not find evidence of, a similar time-dependent drift in HEK cells. Time since rupture was not reported in the studies we reviewed, which makes it difficult to classify this effect. First, between-experiment variability may arise if highly systematic approaches are employed, but these differ between experiments/studies. Any unsystematic deviation cell-to-cell, e.g., due to the time needed to note down cell measurements or adjust compensation circuitry, will lead to within-experiment variability. Next, a correlated means effect could arise, for example, if a systematic approach was followed, if both midpoints were measured in the same cells (consistent with the similar $n_a$ and $n_i$ shown in Table S1), and if the time between activation and inactivation protocols was short relative to the time needed to set up. Because of these uncertainties, we list "time since rupture" as only a possible effect in all three columns of Table 1. The magnitude of these three effects is impossible to determine from our data, but we might estimate an upper bound of 30 min between rupture and measurement, corresponding to 15 mV.

### Other factors

Stretch induced by deliberate pressure applied to oocytes has been shown to shift midpoints of activation by >10 mV (Banderali et al., 2010). If smaller amounts of pressure could be applied *accidentally*, for example by pressure from liquid flow or a badly positioned pipette, could we expect some within-experiment variability as a result? Endogenous currents are known to be present in expression systems, which can interfere with midpoint measurements (Zhang et al., 2022). Use of different cell lines, with different levels of endogenous currents, may cause between-experiment variability, while differing expression levels in each cell could cause within-experiment variability. Culturing conditions and passage number effects could affect channel expression, expression of endogenous currents, or other properties that potentially alter the midpoints (for example, the ability to gain low resistance access), although we know of no data to indicate the size or scale or such an effect. Finally, several factors, including channel glycosylation and phosphorylation, regulate $I_{Na}$ in cardiomyocytes (Marionneau and Abriel, 2015; Daimi et al., 2022). While some of these mechanisms may be highly specialized to cardiomyocytes, we might expect some forms of biological regulation even in cells non-natively expressing sodium channels, which could cause any type of variability depending on how the mechanisms themselves vary.

### Implications

The existence of substantial variability, whether biological or technical, has implications for experimental design and interpretation, for combining studies on $I_{Na}$ in a theoretical, computational, or machine-learning framework, for future reporting on electrophysiology experiments, and for our general understanding of $I_{Na}$.

Firstly, for studies into effects of mutations, drugs, or any other factors affecting $I_{Na}$, our observations underscore the

already well-established need for recent control measurements accompanying every test group. A rule of thumb may be that our confidence in observed differences should increase when studies more closely approach a "paired sample" design. For example, measurements of drug effects where a before and after is available in each cell might be trusted with smaller sample sizes than measurements of mutant versus wild-type cells performed on the same day by the same experimenter, and when more things change (longer time between measurements, change in experimenter or patch-clamp "rig," new batch of solutions, etc.) we should begin to expect the within-study between-experiment variability of Fig. 3, E and F, and adjust our confidence and sample sizes accordingly. In general, the ~10 mV range of between-experiment within-study values and the even wider 40 mV range between experiments in different studies suggest we may need to add some "safety factor" in experiment design and use larger sample sizes and lower P values in significance tests than commonly appreciated.

For cases where pairing is not possible, the case is less clear. For example, how do we interpret studies measuring the "canonical" electrophysiology in a particular cell type and species (e.g., Sakakibara et al., 1992; Sakakibara et al., 1993) or measurements in patient-derived stem cells?

Secondly, the strong correlation between the *mean* midpoints of activation and inactivation ($\mu_a$ and $\mu_i$) suggests a correlation between the *individual* midpoints per cell ($V_a$ and $V_i$), and this is further corroborated by the strong relationship between $\mu_a$ and $V_{peak}$ in Fig. 5 E. As a result, detailed studies measuring individual features of $I_{Na}$ (activation, fast and slow inactivation, deactivation, late component, etc.) *in isolation* risk missing physiologically important relationships between those features, and a full picture of $I_{Na}$ based on disparate recordings could suffer from a "failure of averaging" (Golowasch et al., 2002). An emerging technology that could help address this issue is the use of short, information-rich voltage protocols, which target multiple features of ionic currents at once (Beattie et al., 2018)—although these protocols are themselves derived from preliminary modelling work on conventional protocol data. If using conventional protocols, a good start would be to report the individual $V_a$ and $V_i$ in a figure similar to Fig. 3.

Thirdly, any attempt at data integration, that is combining data from different sources through mechanistic modelling, machine learning, or meta-analysis, should take into account the wide between-experiment variability of Fig. 2 A, the within-experiment variability of Fig. 2 C, and the correlations of Fig. 3 A. Creators of mechanistic (e.g., Clancy and Rudy, 2002) and statistical (or machine-learning, e.g., Clerx et al., 2018) $I_{Na}$ models have long recognized the difficulty of combining seemingly conflicting data from different sources. The results shown here may go some way toward explaining these difficulties and suggest that approaches incorporating at least a degree of variability (Kernik et al., 2019) or uncertainty (Pathmanathan et al., 2015) are required. The distinctions between sources of variability are important for computational work: if technical artefacts explain the majority of the results above, the variation should not be taken as a characterization of cell-to-cell variation for simulation studies of physiological variability. Additionally,

the results suggest that incorporating *changes* (e.g., shifts in midpoints measured against controls) in a baseline model is preferable to including new absolute values, and that—unless confounders are known and reported (see below)—targeted studies into effects of, e.g., subunit types are preferred to meta-analyses as in Figs. 4 and 5.

Fourthly, as it is possible that most of the variability is due to experimental factors that were not reported but known or easily measurable at the time, this study re-emphasizes the need for greater sharing of data and metadata, already acknowledged in standards such as MICEE (Quinn et al., 2011). For example, the data set used here was created by extracting only six core numbers per experiment from each study, while thousands of data points were recorded originally for each *cell*. Taking advantage of modern data-sharing techniques will allow future researchers to perform far more in-depth analyses. An exciting new opportunity for metadata is offered by recent USB-connected patch-clamp amplifiers, which can automatically store the applied voltage protocols, series resistance, cell capacitance, correction and compensation settings, and more, all in the same file as the measured currents. This has the potential to greatly enhance what future modelling, machine-learning, and meta-analyses can do, particularly if (1) a strong data and metadata-sharing culture is established and (2) either open-source or open-but-proprietary file formats are used (e.g., the HEKA PatchMaster format). The difficulties posed by between-experiment variability for data integration are likely to also be relevant to funders, publishers, and universities, who are increasingly trying to move away from treating papers as insular results, instead trying to build strongly linked networks of reusable resources.

Finally, even when confounding variables are controlled in a single-lab multi-experiment study, a between-experiment variability of 7–11 mV remains (Tan et al., 2005; Kapplinger et al., 2015). It is a fascinating question whether this is due to as-of-yet unknown processes native to the cell, a more mundane drift in experimental conditions, or even a result of limited sample size.

## Conclusion and future directions

We reviewed 157 reported mean midpoints of activation ($\mu_a$) and 165 reported mean midpoints of inactivation ($\mu_i$), gathered from 117 publications and found both within-experiment and between-experiment variability. Within experiments, the median standard deviation was 4.0 mV ($\sigma_a$) or 3.6 mV ($\sigma_i$), equivalent to 5th-to-95th percentile ranges of 13 and 12 mV, respectively. Between experiments, values varied over a range of 39 mV ($\mu_a$) or 51 mV ($\mu_i$), with 5th-to-95th percentile ranges of 29 and 36 mV. Grouping by the known and reported biological confounders, α-subunit, β1 co-expression, and cell type did not fully explain this variability. In the 150 experiments providing both $\mu_a$ and $\mu_i$, we found a significant correlation with a slope almost equal to 1, hinting at some unknown factor(s) affecting both midpoints equally. While it is tempting to look for biological causes of such variability, several experimental confounders exist, which means no such conclusions can be drawn from an analysis of the published literature. These results show that care must be taken in situations where paired experiments are not

possible or when data about different facets of channel behavior are taken from different studies (e.g., in modelling). They also highlight the need to take full advantage of new data recording and sharing opportunities, far beyond the scope of traditional methods sections, so that future meta-analyses may untangle the different possible sources of variability. We conclude that a larger-than-hitherto-reported variability exists in the midpoints of activation and inactivation of $I_{Na}$ and that the mean midpoints are highly correlated. And while the available evidence leaves room for the existence of cell-to-cell variability in the voltage dependence of $I_{Na}$ (with some regulatory mechanism maintaining a certain difference between the two), a simpler explanation at this point is that unreported experimental confounders give rise to the observed variability.

### Data availability

A database containing all data used in this study, along with code to generate all figures, tables, and numbers in the manuscript, is available for download from https://github.com/CardiacModelling/ina-midpoints and permanently archived at https://doi.org/10.5281/zenodo.15697497. The main data (midpoints, standard deviations, cell counts, and references) is provided in tabular form in Table S3.

### Acknowledgments

David A. Eisner served as editor.

This work was supported by the Wellcome Trust (grant no. 212203/Z/18/Z), the Biotechnology and Biological Sciences Research Council (grant number BB/P010008/1), and the Netherlands CardioVascular Research Initiative (grant nos. CVON2017-13 VIGILANCE and CVON2018B030 PREDICT2). G.R. Mirams and M. Clerx acknowledge support from the Wellcome Trust via a Wellcome Trust Senior Research Fellowship to G.R. Mirams. P.G.A. Volders received funding from the Netherlands Cardio-Vascular Research Initiative and the Health Foundation Limburg. This research was funded in whole, or in part, by the Wellcome Trust [212203/Z/18/Z].

For the purpose of open access, the author has applied a CC-BY public copyright license to any author-accepted manuscript version arising from this submission. Open Access funding provided by University of Nottingham.

Author contributions: M. Clerx: Conceptualization, data curation, formal analysis, investigation, methodology, software, validation, visualization, and writing—original draft, review, and editing. P.G.A. Volders: conceptualization, funding acquisition, investigation, project administration, supervision, and writing—review and editing. G.R. Mirams: conceptualization, formal analysis, methodology, supervision, visualization, and writing—original draft, review, and editing.

Disclosures: The authors declare no competing interests exist.

Submitted: 6 June 2024

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

# Supplemental material

Figure S1. **Correlations between reported experimental factors and mean midpoints of activation (μ$_a$) and inactivation (μ$_i$), indicated by an orange linear regression line.** The number of data points and the coefficient of determination are shown above each panel. Factors shown are the steepness of the activation curve (k$_a$); the approximate magnitude of a "representative" current, if one was shown; the holding potential in the activation protocol (V$_{hold,a}$) and inactivation protocol (V$_{hold,i}$); and external and internal concentrations of sodium, calcium, and chloride. The internal calcium concentrations shown were calculated using Maxchelator (Bers et al, 2010). This figure has two major caveats: (1) the variable on the x axis is not the only one varied between experiments, and since experimental design choices are often inherited from previous work, we can also expect them to show some correlation (e.g., copying both a holding potential and bath/pipette solutions from the same seminal work); (2) some choices are so common that the "groups" on the x axis are very small, making correlations more spurious. For example, only 27 experiments used a nonzero [Ca$^{2+}$]$_i$.

Provided online are Table S1, Table S2, and Table S3. Table S1 shows all the reviewed studies containing more than one experiment. Table S2 shows a "histogram" view of the difference in cell counts ($|n_a - n_i|$) and how often each was encountered. Table S3 shows all the experiments reviewed in this manuscript.

