## [Peer Review File · The Journal of General Physiology]

Variability in reported midpoints of (in)activation of cardiac INa

Michael Clerx, Paul Volders, and Gary Mirams

Corresponding Author(s): Michael Clerx, University of Nottingham

Review Timeline:

Submission Date:	June 6, 2024
Editorial Decision:	July 3, 2024
Revision Received:	May 23, 2025
Editorial Decision:	June 12, 2025
Revision Received:	June 19, 2025

Editor: David Eisner

Transaction Report:

DOI: <https://doi.org/10.1085/jgp.202413621>

July 3, 2024

Dr. Michael Clerx
University of Nottingham
Mathematical Sciences
University Park Campus
Nottingham NG7 2RD
United Kingdom

Re: 202413621

Dear Dr. Clerx,

Thank you for submitting your manuscript, entitled "Variability in reported midpoints of (in)activation of cardiac INa" to JGP. Your manuscript has now been seen by 3 reviewers, whose comments are appended below. You will see that the reviewers were very enthusiastic about the study and its potential impact and raised only minor concerns that should nevertheless be addressed prior to further consideration of the manuscript at JGP. In particular, please note the suggestions from the reviewers to discuss, preferably in a quantitative manner, the relative contributions of the various factors to the spread of inter-experiment values. In addition, please clarify the meaning of "correlated versus uncorrelated in both the abstract and Table 1.

We hope that you will be able to submit a revised manuscript that addresses these points, which we believe will pose no problems, and which may be re-reviewed. In addition, please do not hesitate to contact me (via the editorial office) if you feel that a discussion of the reviewers' and editors' comments would be helpful.

Please submit your revised manuscript via the link below, along with a point-by-point letter that details your response to the reviewers' and editors' comments, as well as a copy of the text with alterations highlighted (boldfaced or underlined). If the article is eventually accepted, it would include a 'revised date' as well as submitted and accepted dates. If we do not receive the revised manuscript within one year, we will regard the article as having been withdrawn. We would be willing to receive a revision of the manuscript at a later time, but the manuscript will then be treated as a new submission, with a new manuscript number.

Please pay particular attention to recent changes to our instructions to authors in the following sections: Data presentation, Blinding and randomization and Statistical analysis, under Materials and Methods, as shown here: <https://rupress.org/jgp/pages/submission-guidelines#prepare>. Re-review will be contingent on inclusion of the required information (including for data added during revision) and demonstration of the experimental reproducibility of the results. Also, To improve the reproducibility of published content, we have partnered with SciScore. Authors are prompted in eJP to copy and paste the Materials and Methods section of their manuscript for a SciScore assessment when submitting their revised manuscript. Authors are encouraged (not required) to further revise their Materials and Methods if the SciScore is below 4. More information can be found here: <https://rupress.org/jgp/pages/submission-guidelines#sciscore>.

Please note, JGP now requires authors to submit Source Data used to generate figures containing gels and Western blots with all revised manuscripts (when applicable). This Source Data consists of fully uncropped and unprocessed images for each gel/blot displayed in the main and supplemental figures. If your paper includes cropped gel and/or blot images, please be sure to provide one Source Data file for each figure that contains gels and/or blots along with your revised manuscript files. File names for Source Data figures should be alphanumeric without any spaces or special characters (i.e., SourceDataF#, where F# refers to the associated main figure number or SourceDataFS# for those associated with Supplementary figures). The lanes of the gels/blots should be labeled as they are in the associated figure, the place where cropping was applied should be marked (with a box), and molecular weight/size standards should be labeled wherever possible. Source Data files will be made available to reviewers during evaluation of revised manuscripts and, if your paper is eventually published in JGP, the files will be directly linked to specific figures in the published article.

Source Data Figures should be provided as individual PDF files (one file per figure). Authors should endeavor to retain a minimum resolution of 300 dpi or pixels per inch. Please review our instructions for export from Photoshop, Illustrator, and PowerPoint here: <https://rupress.org/jgp/pages/submission-guidelines#revised>

Whilst you are revising your manuscript, we ask that you consider whether you have any artwork that might be suitable for the cover of JGP. Microscopy images are particularly good for cover artwork, but other types of image can be very effective, so we encourage you to be creative. Please don't restrict yourself to images from the paper; an image that is relevant to the work described would be just as suitable. Images should be a minimum resolution of 300 dpi. To see recent examples, visit the following page and click on 'Show covers? Yes': <https://jgp.rupress.org/content/by/year>

Thank you for submitting your interesting research to JGP.

Please submit your revised manuscript, and any associated files, via this link:
Link Not Available

Sincerely,

David Eisner, D. Phil
On behalf of Journal of General Physiology

Journal of General Physiology's mission is to publish mechanistic and quantitative molecular and cellular physiology of the highest quality; to provide a best-in-class author experience; and to nurture future generations of independent researchers.

Reviewer #1 (Comments to the Authors):

This is a fascinating study exploring in detail and quantifying the variability in two critical electrophysiological parameters: the midpoints of activation and inactivation of the cardiac fast sodium current, INa. It is well documented that electrically active cells, such as cardiomyocytes, are known for their variability in size, shape, and electrical activity. The question of whether this variability extends to their ionic currents has been discussed as a notable issue in earlier studies and the potential impact on cellular electrical variability -might be worth mentioning some of them including: A visual comparison and noted likewise shift of h and m (and recovery from inactivation) in the supplemental figures 1 and 2 in <https://doi.org/10.1161/hc1002.105183> as well as the strong variability in measurements of INa and other currents in iPSC-CM in DOI: 10.1113/JP277724

One of the most notable contributions of this study is the decomposition of between-experiment variability into correlated and uncorrelated components. The study finds that the correlated component is substantially larger, influencing both activation and inactivation midpoints almost equally. This insight is vital for the scientific community, as it underscores the importance of considering experimental context and inherent biological variability when interpreting electrophysiological data.

The authors delve into potential biological and methodological factors that could explain the observed variability, categorizing each as within-experiment or correlated and uncorrelated between-experiment factors. This systematic approach is a key piece of the study in identifying sources of variability but also provides a framework for future research to refine experimental designs and improve the reliability of measurements.

This is an elegant highly readable study. I think it could be improved by noting the potential impact of such variability on the interpretation of studies and our understanding of the fundamental mechanisms of h and m on cellular emergent behaviors - for example, what does it mean when a small shift in one of these parameters is linked to inherited or acquired disease when the change is within the range of normal? How should the perspective presented here affect the way we interpret such effects?

Reviewer #2 (Comments to the Authors):

Clerx et al., performed a literature survey and reviewed the variability in reported V1/2s of activation and inactivation of the cardiac sodium channels expressed in HEK or CHO cells. They examined the distribution of V1/2s among 174 experiments from 120 published studies. They showed that the measurements of V1/2s of activation and inactivation are highly correlated among different experiments, and the variability cannot be explained by sample size or expression of different alpha isoforms or the beta1 subunit. Overall, the review is well-written and insightful. However, some aspects can still be improved to make the study more informative.

Major comments:

1. Although the authors discussed many different factors that can contribute to the variability among different studies (e.g. LJPs, voltage protocol differences, voltage control), they did not examine any of these aspects with the data they curated. Some of these factors can be tested using the data in the literature. For example, does the peak current amplitude, slope of the activation curve (steep slope sometimes is indicative of a voltage control problem), or [Na] in external solution affect the V1/2 measured? Or does the holding potential/time in the voltage protocol or the source or passage number of HEK/CHO cells affect the V1/2? Testing some of these hypotheses will further inform the origination of experimental variabilities.

2. Could the authors specify how the V1/2s were derived from the studies? Were they fitted from the curves published in previous studies, or were they directly extracted from the reported V1/2? If the latter, could the curve fitting process also contribute to the variability?

Minor comments:

1. The terms "correlated" and "uncorrelated" need to be better defined/explained in the abstract.

Reviewer #3 (Comments to the Authors):

In this meta-analysis, the authors examined the midpoints of activation and inactivation curves of the cardiac voltage gated sodium current (INa) reported in a set of 120 studies that investigated this current in HEK or CHO cells expressing wild-type

human cardiac sodium channels.

They found that these reported midpoints exhibit a very large variability, not only between different investigator groups but also within different series in individual laboratories. Moreover, the authors observed a significant correlation between the midpoint of activation and the midpoint of inactivation. In their Discussion, the authors then debate about the possible causes of this variability.

These findings bear important consequences for electrophysiologists and highlight the notion that care must be taken when comparing results from different studies.

The article is well presented, has great didactic potential, and represents an important work for electrophysiologists studying ion currents with the patch clamp technique. It is worth to be read by all electrophysiologists at all career levels, from the graduate student to the established investigator.

I have the following specific suggestions and comments:

The didactic impact could be further strengthened by explaining in the introduction why INa is important for cardiac function. Moreover, the standard protocols to study INa activation and inactivation could be presented in more detail, as well as the ways in which the recorded currents are typically analyzed (i.e., identifying peaks, normalization, fitting with specific (e.g., Boltzmann) functions, etc.).

As further causes of variability (Discussion and Table 1), the authors could consider uncontrollable redox (half-battery) potentials that may arise on the electrodes depending on their level of chlorination. Moreover, the exact composition of the intracellular (pipette) and extracellular solutions (ion concentrations, calcium buffers, presence or absence of drugs blocking other channels) may also affect the results. Furthermore, in expression systems, there is typically an enormous variability in the number of channels expressed, and hence on total INa. Researchers may prefer to include only experiments with large currents, which may introduce bias. Finally, the exact culture conditions as well as drifts in cellular phenotype after repeated trypsinization passages of the HEK or CHO cells may also play an important role.

Minor comments:

In the abstract, the words "In this brief review..." are misleading, as this article is rather a meta-analysis.

Figure 1B is not mentioned in the Results section. The x-axes of panels A and B could be aligned to allow a direct visual comparison.

Discussion, series resistance: Series resistance depends on pipette properties (e.g., tip diameter) rather than on the quality of the seal.

Point-by-point reply, “Variability in reported midpoints of (in)activation of cardiac I_{Na} ”

We thank all reviewers for their encouraging comments and helpful suggestions.

Reviewer 1's comments

“This is a fascinating study exploring in detail and quantifying the variability in two critical electrophysiological parameters: the midpoints of activation and inactivation of the cardiac fast sodium current, I_{Na} . It is well documented that electrically active cells, such as cardiomyocytes, are known for their variability in size, shape, and electrical activity. The question of whether this variability extends to their ionic currents has been discussed as a notable issue in earlier studies and the potential impact on cellular electrical variability - might be worth mentioning some of them including: A visual comparison and noted likewise shift of h and m (and recovery from inactivation) in the supplemental figures 1 and 2 in <https://doi.org/10.1161/hc1002.105183> as well as the strong variability in measurements of I_{Na} and other currents in iPSC-CM in <https://doi.org/10.1113/JP277724>”

Thank you for these kind words and for pointing out these studies. Both are now mentioned in the updated “Implications” section, on lines 388 and 391.

One of the most notable contributions of this study is the decomposition of between-experiment variability into correlated and uncorrelated components. The study finds that the correlated component is substantially larger, influencing both activation and inactivation midpoints almost equally. This insight is vital for the scientific community, as it underscores the importance of considering experimental context and inherent biological variability when interpreting electrophysiological data.

The authors delve into potential biological and methodological factors that could explain the observed variability, categorizing each as within-experiment or correlated and uncorrelated between-experiment factors. This systematic approach is a key piece of the study in identifying sources of variability but also provides a framework for future research to refine experimental designs and improve the reliability of measurements.

This is an elegant highly readable study. I think it could be improved by noting the potential impact of such variability on the interpretation of studies and our understanding of the fundamental mechanisms of h and m on cellular emergent behaviors

Thank you for this suggestion. In response we have restructured, rewritten, and strengthened the “Implications” section at the end of the discussion, on lines 355–398.

for example, what does it mean when a small shift in one of these parameters is linked to inherited or acquired disease when the change is within the range of normal? How should the perspective presented here affect the way we interpret such effects?

In the updated “Implications” section, we argue that (1) as long as experiments are sufficiently “paired” such small but *relative* effects may actually be quite knowable, compared to the absolute baseline values, but that (2) given the within-study between-experiment variability, which ranges over about 10mV, we should still treat them with some caution and perhaps start using larger sample sizes and smaller p-values. Similarly, when interpreting existing measurements the asterisks indicating significance may need to be taken with a pinch of salt. Although we hesitate to touch on disease in the manuscript (limiting the scope of this work to basic cell electrophysiology), particularly important small changes would benefit from corroborative evidence in other measurement types, e.g. measurements of upstroke velocity or propagation speed in transgenic mice.

Reviewer #2 (Comments to the Authors):

Clerx et al., performed a literature survey and reviewed the variability in reported V1/2s of activation and inactivation of the cardiac sodium channels expressed in HEK or CHO cells. They examined the distribution of V1/2s among 174 experiments from 120 published studies. They showed that the measurements of V1/2s of activation and inactivation are highly correlated among different experiments, and the variability cannot be explained by sample size or expression of different alpha isoforms or the beta1 subunit. Overall, the review is well-written and insightful. However, some aspects can still be improved to make the study more informative.

Major comments:

1. Although the authors discussed many different factors that can contribute to the variability among different studies (e.g. LJPs, voltage protocol differences, voltage control), they did not examine any of these aspects with the data they curated. Some of these factors can be tested using the data in the literature. For example, does the peak current amplitude, slope of the activation curve (steep slope sometimes is

indicative of a voltage control problem), or [Na] in external solution affect the V_{1/2} measured? Or does the holding potential/time in the voltage protocol or the source or passage number of HEK/CHO cells affect the V_{1/2}? Testing some of these hypotheses will further inform the origination of experimental variabilities.

Thank you for your kind assessment and for these suggestions. In response, we revisited the studies included in our study and noted these and other factors where possible. This is reflected in a new section “Experimental variability” on line 210–227 of the Results, a new Figure 5 in the manuscript, and a new Supplemental figure S1 investigating linear correlations between the mean midpoints and various experimental factors, including [Na]_e. The new results are also discussed in the relevant sections of the “Experimental sources of variability” section in the Discussion.

Interestingly, the slope of the activation curve was not a strong predictor for the mean midpoint of activation (see the new Figure 5D) — although we recently showed that such a correlation at single cell level may disappear when results are averaged (<https://doi.org/10.1101/2024.07.23.604780>). The peak current amplitude was very rarely given in the pA needed to assess voltage control, and only occasionally in pA/pF. Many papers did show a “representative” I_{Na} trace in pA, which we attempted to use as a rough estimate of what the typical peak I_{Na} may have been - although it is likely these “representative” currents were much larger (and hence less noisy and prettier) than the currents used in analysis. This peak representative current did not correlate with μ_a , as shown in the new Supplementary Figure S1.

We have updated our discussion of the LJP on lines 211–214 of Results and lines 272–278 of the Discussion. Remarkably few studies mentioned correcting for the LJP, and all reported midpoints at the lower (more polarised) end of the spectrum, suggesting that the LJPs of the various solutions used could play a part in between-experiment variability.

Holding potential had a relatively strong effect on both midpoints (new Figure 5B and 5C), but cycle or holding times were not provided frequently enough for analysis (see also discussion on lines. Similarly, the source and passage number were not provided frequently enough for analysis. We do discuss these factors qualitatively on lines 345–348.

Finally, we note that one implication of our results in Figures 2 and 3 is that this kind of meta-analysis of pooled studies is not well-suited to observe quantitative or qualitative effects, as several confounding factors may obscure results that would be clearly visible in a targeted study from a single lab. We also suspect that multiple experimental choices are often concomitantly

inherited from previous work, which could create some spurious relationships in the data.

2. Could the authors specify how the $V_{1/2}$ s were derived from the studies? Were they fitted from the curves published in previous studies, or were they directly extracted from the reported $V_{1/2}$? If the latter, could the curve fitting process also contribute to the variability?

We only included studies that reported μ_a and/or μ_i numerically, and took these numbers directly from the reviewed manuscripts without performing any new curve fitting or other analysis. This is now stated more clearly, at the end of the Background section (lines 90–91) and in the Methods section (Lines 126–128). In addition, we have extended our discussion of the experimental analysis methods on lines 307–310, and now list curve fitting separately in Table 1 (as “Analysis method”).

Minor comments:

1. The terms "correlated" and "uncorrelated" need to be better defined/explained in the abstract.

We have updated the abstract on lines 7–9 to provide a clearer definition.

Reviewer #3 (Comments to the Authors):

In this meta-analysis, the authors examined the midpoints of activation and inactivation curves of the cardiac voltage gated sodium current (INa) reported in a set of 120 studies that investigated this current in HEK or CHO cells expressing wild-type human cardiac sodium channels.

They found that these reported midpoints exhibit a very large variability, not only between different investigator groups but also within different series in individual laboratories. Moreover, the authors observed a significant correlation between the midpoint of activation and the midpoint of inactivation. In their Discussion, the authors then debate about the possible causes of this variability.

These findings bear important consequences for electrophysiologists and highlight the notion that care must be taken when comparing results from different studies.

The article is well presented, has great didactic potential, and represents an important work for electrophysiologists studying ion currents with the patch clamp technique. It is worth to be read by all electrophysiologists at all career levels, from the graduate student to the established investigator.

I have the following specific suggestions and comments:

The didactic impact could be further strengthened by explaining in the introduction why I_{Na} is important for cardiac function. Moreover, the standard protocols to study I_{Na} activation and inactivation could be presented in more detail, as well as the ways in which the recorded currents are typically analyzed (i.e., identifying peaks, normalization, fitting with specific (e.g., Boltzmann) functions, etc.).

Thank you for your very kind words and for this suggestion. In response, we have added a Background section (including a new Figure 1) which gives a very brief overview of I_{Na} 's role in cardiac function, and the experiments and analysis commonly used to estimate μ_a and μ_i . In the text we follow Hanck & Sheets 1992 (<https://doi.org/10.1152/ajpheart.1992.262.4.H1197>) in (A) assuming a rate-limiting step described by a Boltzmann distribution, rather than a Hodgkin-Huxley model which would add a third power to our Equation 1; and (2) in suggesting that activation experiments be analysed by fitting to current directly, instead of first calculating conductance by dividing through ($V_{test} - E$) as this partially avoids numerical issues for voltages near the reversal potential. We appreciate there is more to be said here, e.g. exactly how peaks are identified in noisy data, analysis near $V=E$, what to do if the protocol does not include the extremes of either Boltzmann curve, advantages of low temperatures or non-physiological solutions with $E=0$, use of cesium etc. — we believe such a paper would be a valuable addition to the literature but hope that an experienced I_{Na} experimenter could write such a paper more informatively and authoritatively. To avoid moving too far from the original goal of this study we have attempted to keep this section very brief and refer to a number of “further reading” papers at the section's end.

As further causes of variability (Discussion and Table 1), the authors could consider uncontrollable redox (half-battery) potentials that may arise on the electrodes depending on their level of chlorination. Moreover, the exact composition of the intracellular (pipette) and extracellular solutions (ion concentrations, calcium buffers, presence or absence of drugs blocking other channels) may also affect the results.

Thank you for these suggestions, we have added redox potentials and bath & pipette solutions to Table 1 and discuss these on lines 279–284 and 311–316. We also revisited all reviewed papers to gather data on the exact solutions used, holding potentials, and other experimental choices. Internal calcium was nominally zero in most experiments, but for the few reports we found of internal calcium and buffers, we attempted to calculate free $[Ca^{2+}]_i$ and look for any effects. The result is shown in supplemental figure S1 but is presumably too clouded by confounding factors to show any effect. As all experiments were in HEK or CHO, we saw no reports of groups adding any drugs or blockers.

Furthermore, in expression systems, there is typically an enormous variability in the number of channels expressed, and hence on total I_{Na} . Researchers may prefer to include only experiments with large currents, which may introduce bias.

In an attempt to address this, we revisited all studies to look for mean peak I_{Na} values, but found these were rarely given in A/F, and even more rarely in the pA needed to assess this. We did note the size of “representative currents” shown in many manuscripts, but saw no correlation between this and the midpoint of activation. This is now shown in Supplemental Figure S1.

Finally, the exact culture conditions as well as drifts in cellular phenotype after repeated trypsinization passages of the HEK or CHO cells may also play an important role.

This is now discussed on lines 345–348, but unfortunately we were unable to find enough information in the reviewed studies to make a quantitative assessment.

Minor comments:

In the abstract, the words "In this brief review..." are misleading, as this article is rather a meta-analysis.

This has been updated on line 3.

Figure 1B is not mentioned in the Results section.

This is now mentioned on line 164.

The x-axes of panels A and B could be aligned to allow a direct visual comparison.

Figure 1 has been updated so that panels A and B share an X-axis.

Discussion, series resistance: Series resistance depends on pipette properties (e.g., tip diameter) rather than on the quality of the seal.

We have removed this part of the sentence.

June 17, 2025

Dr. Michael Clerx
University of Nottingham
Mathematical Sciences
University Park Campus
Nottingham NG7 2RD
United Kingdom

Re: 202413621R1

Dear Dr. Clerx,

I am pleased to let you know that your manuscript, entitled "Variability in reported midpoints of (in)activation of cardiac INa" is scientifically acceptable for publication in Journal of General Physiology. Formal acceptance will follow when it is modified in accordance with the referees' remarks and our editorial policies.

Please note items that need attention are listed at the bottom of this email under 'manuscript formatting checklist'. Please also be sure to include a letter addressing the reviewers' comments point-by-point (if applicable) and a copy of the text with alterations highlighted (boldfaced or underlined). Your manuscript should be a double-spaced MS Word file and include editable tables, if appropriate.

Lastly, JGP requires a data availability statement for all research article submissions. These statements will be published in the article directly above the Acknowledgments. The statement should address all data underlying the research presented in the manuscript. Please visit the JGP instructions for authors for guidelines and examples of statements at <https://rupress.org/jgp/pages/editorial-policies#data-availability-statement>.

Please submit your final files via this link:
Link Not Available

Thank you for choosing to publish your research in JGP and please feel free to contact me with any questions.

Sincerely,

David Eisner, D. Phil
On behalf of Journal of General Physiology

Journal of General Physiology's mission is to publish mechanistic and quantitative molecular and cellular physiology of the highest quality; to provide a best in class author experience; and to nurture future generations of independent researchers.

Manuscript formatting checklist:

- MS Word document of text needed (including editable tables)
 - MS Word document of supplemental text needed, if applicable (including figure legends and editable tables)
 - Brief Statement describing supplementary information needed, if applicable (in subsection at end of Materials & Methods)
 - Please include a data availability statement preceding the Acknowledgments section. Please see <https://rupress.org/jgp/pages/editorial-policies#data-availability-statement>
 - Figures created at sufficient resolution and in acceptable format (including supplemental if applicable). If working in Illustrator, we prefer .ai or .eps file format. If working in Photoshop please use 600dpi/1000dpi .tiff or .psd file format. Minimum resolution at estimated print size: Minimum resolution for all figures is 600 dpi. For figures that contain both photographs and line art or text, 600 dpi is highly recommended. Figures containing only black and white elements (line art, no color, and no gray) should be 1,000 dpi. Maximum figure size is 7 in wide x 9 in high (17.5 x 22.8 cm) at the correct resolution. <https://jgp.rupress.org/fig-vid-guidelines>
 - Supplemental figures, if any, conforming to same guidelines as manuscript figures (noted above)
 - If images resemble one from a prior publications, the author must seek permissions (to reproduce or adapt) from the original publisher. [You can resubmit your paper while waiting to hear back from the original publisher but please keep us updated]
 - All authors must complete a disclosure form prior to acceptance. A link to complete the form has been sent to all coauthors. Please provide the editorial office with updated email addresses if necessary
-

Reviewer #1 (Comments to the Authors):

Authors have provided a detailed and thorough revision. This is a thrilling study - i look forward to sharing it.

Reviewer #2 (Comments to the Authors):

The authors have addressed my comments/concerns adequately. The revised manuscript has been much improved.

Reviewer #3 (Comments to the Authors):

The authors have addressed my comments. Specifically, in their revision, the authors included a new section in the Introduction explaining why INa is important for cardiac function and detailing the standard protocols to study and analyze INa activation and inactivation. This increases the didactic impact of the article. Furthermore, they elaborated on further causes of variability in the midpoints of activation and inactivation, and conducted additional analyses.

Thank you for this important article!